# Neural Task Graph Execution

## Abstract

In order to develop a scalable multi-task reinforcement learning (RL) agent that is able to execute many complex tasks, this paper introduces a new RL problem where the agent is required to execute a given task graph which describes a set of subtasks and dependencies among them. Unlike existing approaches which explicitly describe what the agent should do, our problem only describes properties of subtasks and relationships between them, which requires the agent to perform a complex reasoning to find the optimal subtask to execute. To solve this problem, we propose a neural task graph solver (NTS) which encodes the task graph using a recursive neural network. To overcome the difficulty of training, we propose a novel non-parametric gradient-based policy that performs back-propagation over a differentiable form of the task graph to compute the influence of each subtask on the other subtasks. Our NTS is pre-trained to approximate the proposed gradient-based policy and fine-tuned through actor-critic method. The experimental results on a 2D visual domain show that our method to pre-train from the gradient-based policy significantly improves the performance of NTS. We also demonstrate that our agent can perform a complex reasoning to find the optimal way of executing the task graph and generalize well to unseen task graphs. In addition, we compare our agent with a Monte-Carlo Tree Search (MCTS) method showing that our method is much more efficient than MCTS, and the performance of our agent can be further improved by combining with MCTS. The demo video is available at https://youtu.be/e_ZXVS5VutM.

## 1 Introduction

Developing the ability to execute many different tasks depending on given task descriptions and generalize over unseen task descriptions is an important problem for building scalable reinforcement learning (RL) agents. Recently, there have been a few attempts to define and solve different forms of task descriptions such as natural language (Oh et al., 2017; Yu et al., 2017) or formal language (Denil et al., 2017). However, most of the prior work has focused on task descriptions which explicitly specify what the agent should do, which may not be easy in real-world applications.

Suppose that we want to ask a household robot to make a breakfast in an hour. A breakfast meal may be served with different combinations of dishes, each of which takes a different cost (e.g, time) and gives a different amount of reward (e.g. user satisfaction) depending on the user preferences. In addition, there can be complex dependencies between subtasks. For example, a bread should be sliced before toasted, or an omelette and an egg sandwich cannot be made together if there is only one egg left. Due to such complex dependencies as well as different rewards and costs, it is often difficult for human users to manually find the best combination and sequence of subtasks and explicitly describe what the agent should do (e.g., "fry an egg and toast a bread"). Instead, it is more desirable for the agent to figure out the optimal sequence of subtasks that gives the maximum reward within a time budget just from properties and depedencies of subtasks.

The goal of this paper is to define and solve such a problem, which we call *task graph execution*, where the agent is required to execute the given *task graph* in an optimal way as illustrated in Figure 1. More specifically, a task graph consists of subtasks, corresponding rewards, and dependencies among subtasks in the sum-of-product (SoP) form. The SoP is expressive enough to represent any logical expressions and subsume many existing forms (e.g., sequential instructions (Oh et al., 2017)). This allows us to define many complex tasks in a principled way and train the agent to find the optimal way of executing such tasks. The task graph execution problem is very challenging because the agent should consider the long-term effect of each subtask due to deep dependencies among subtasks. In addition, the agent is required to generalize over unseen task graphs during evaluation.

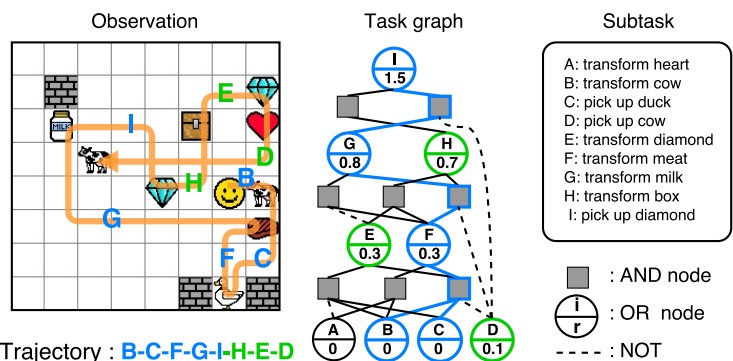

Figure 1: Example task and our agent's trajectory. The agent is required to execute subtasks in the optimal order to maximize the reward within a time limit. The task graph describes subtasks with the corresponding rewards (e.g. subtask `F` gives 0.3 reward) and dependencies between subtasks through `AND` and `OR` nodes. For instance, in order to execute subtask `F`, the agent needs to satisfy its precondition: `OR(AND(A, B), AND(B, C, NOT(D))))`. In this example, our agent learned not to execute `D` at the beginning even though `D` gives an immediate reward, because executing `D` makes `I` not executable due to `NOT` operation, which gives the largest reward. Thus, our agent chose to satisfy the preconditions of `I` and execute it (blue), and chose to execute remaining subtasks later (green).

To solve the problem, we propose a new deep RL architecture, called *neural task graph solver* (NTS), which encodes a task graph using a recursive-reverse-recursive neural network (R3NN) (Parisotto et al., 2016) to consider the long-term effect of each subtask. To address the difficulty of learning, we propose to pre-train the NTS to approximate our novel non-parametric gradient-based policy called *reward-propagation policy*. The key idea of reward propagation policy is to construct a differentiable representation of the task graph such that taking a gradient over the reward amounts to propagating reward information between related subtasks. Since our reward-propagation policy acts as a good initial policy, we train the NTS to approximate the reward-propagation policy through policy distillation (Rusu et al., 2015; Parisotto et al., 2015) and fine-tune it through actor-critic method (Konda & Tsitsiklis, 1999).

To evaluate our method, we introduce a 2D visual grid-world domain with a set of task graphs that contain diverse types of task dependencies. Our experimental results show that the proposed reward-propagation policy is crucial for training our NTS agent, and our agent outperforms all the baselines. We also provide empirical evidences that our agent implicitly performs a complex reasoning by taking into account long-term task dependencies as well as the cost of executing each subtask from the observation, and it can successfully generalize to unseen and larger task graphs. In addition, we compare our agent with a Monte-Carlo tree search (MCTS) algorithm. The results show that our method is computationally much more efficient than MCTS. Finally, we also show that the performance of our NTS agent can be further improved by combining with MCTS, achieving a near-optimal performance.

## 2 RELATED WORK

**Programmable Agent**   The idea of learning to execute a given program using RL was introduced by *programmable hierarchies of abstract machines* (PHAMs) (Parr & Russell, 1997; Andre & Russell, 2000; 2002). PHAMs specify a partial policy using a set of hierarchical finite state machines, and the agent is required to learn the optimal completion of the partial program. Andreas et al. (2017) explored a different way of specifying a partial policy in the deep RL framework. There have been other approaches that use a program as a form of task description rather than a partial policy in the context of multi-task RL (Oh et al., 2017; Denil et al., 2017). Our work also aims to build a programmable agent in that we describe a task in a form of language and train the agent to execute it. However, most of the prior work has focused on a setting where the program specifies what to do, and the agent just needs to learn how to do. In contrast, our work explores a new form of program, called *task graph* (see Figure 1), which describes properties of several tasks and dependencies between them, and the agent is required to figure out what to do as well as how to do it.

**Program Induction and Synthesis**   Recently, there have been a few attempts to infer a program from examples  (Reed & De Freitas, 2015; Cai et al., 2017; Parisotto et al., 2016). For example,

*neural programmer-interpreter* (NPI) (Reed & De Freitas, 2015) proposed a neural network that infers a program execution trace from an input. Parisotto et al. (2016) also proposed a neural network to synthesize a tree-structured program that transforms an input to an output. Most recently, Xu et al. (2017) extended this idea to RL problems by learning to infer the underlying program from demonstrations. In contrast to this line of work, we focus on the opposite problem: how to optimally execute a given program (i.e., task graph) in RL context.

**Hierarchical Reinforcement Learning**   Many hierarchical RL approaches have been proposed to solve complex decision problems by building multiple levels of temporal abstractions (Sutton et al., 1999; Dietterich, 2000; Precup, 2000; Ghavamzadeh & Mahadevan, 2003; Konidaris & Barto, 2007). By following this idea, we also present a hierarchical RL architecture where the high-level controller focuses on finding the optimal subtask from the task graph and the observation, while the low-level controller focuses on executing the given subtask. In our work, however, we mainly focus on how to train the high-level controller to deal with delayed reward and long-term dependencies between subtasks.

**Planning with Hierarchical Task Network**   One of the most closely related problem to our task graph execution problem is the planning problem considered in hierarchical task network (HTN) approaches (Sacerdoti, 1975; Erol, 1996; Erol et al., 1994; Nau et al., 1999; Castillo et al., 2005) in that HTN approaches also aim to find the optimal way to execute tasks given task dependencies and cost information. However, HTN approaches aim to execute a single goal task, while the goal of our problem is to maximize the cumulative reward without a particular goal task. Thus, the agent in our problem not only needs to consider complex dependencies among different tasks but also needs to infer the cost from the observation. These additional challenges make it difficult to directly apply HTN approaches to solve our problem.

**Motion Planning**   Another related problem to our task graph execution problem is motion planning (MP) problem (Asano et al., 1985; Canny, 1985; 1987; Faverjon & Tournassoud, 1987; Keil & Sack, 1985). Solving MP problem often involves solving graph search problem after reducing or mapping given MP problem to the graph. However, different from our problem, the MP approaches aim to find an optimal path to the goal in the graph while avoiding obstacles similar to HTN approaches.

## 3   THE TASK GRAPH EXECUTION PROBLEM

Let $\mathcal{S}$ be a set of states, $\mathcal{G}$ be a set of task graphs, $\mathcal{A}$ be a set of actions, and $\gamma$ be a discount factor. The *task graph execution* problem is defined as a Markov Decision Process (MDP): $\mathcal{M} = (\mathcal{S}, \mathcal{A}, \mathcal{G}, \mathcal{R}, \gamma)$ where the reward function is defined as $\mathcal{R} : \mathcal{S} \times \mathcal{G} \times \mathcal{A} \to \mathbb{R}$. We assume that the agent has a set of pre-learned *options* (Precup (2000); Stolle & Precup (2002); Sutton et al. (1999)) ($\mathcal{O}$) that performs subtasks by executing one or more primitive actions. More specifically, we define a semi-MDP (SMDP) as $\mathcal{M}' = (\mathcal{S}, \mathcal{O}, \mathcal{G}, \mathcal{R}, \gamma)$. The goal is to learn a multi-task policy $\pi : \mathcal{S} \times \mathcal{G} \to \mathcal{O}$ which chooses the optimal subtask given the current state and the task graph to maximize the cumulative discounted reward $R = \mathbb{E}\left[\sum_{t=0}^{T} \gamma^t r_t\right]$ in an episode, where $r_t$ is the reward at time step $t$.

A task graph $\mathbf{G} \in \mathcal{G}$ consists of subtasks with corresponding rewards and preconditions as illustrated in Figure 1. A *precondition* of a subtask is defined as a logical expression of other subtasks in sum-of-products (SoP) form where multiple AND terms are combined with an OR term (e.g. OR(AND(A, B), AND(B, C, NOT(D)))) in Figure 1). Since SoP can represent any logical expression, we can define complex task dependencies in the form of a task graph.

A subtask is *eligible* if and only if its precondition is satisfied and it has never been executed by the agent. The agent receives the reward associated with the subtask $i$ if and only if the agent executes the subtask $i$ and the subtask $i$ was eligible. We define *subtask completion indicator* $\mathbf{x}_t \in \{0, 1\}^N$ where $x_t^i = 1$ if and only if subtask $i$ has been executed by the agent. We also define *task eligibility vector* $\mathbf{e}_t \in \{0, 1\}^N$ where $e_t^i = 1$ if and only if the precondition of subtask $i$ is satisfied. These two vectors $\mathbf{x}_t, \mathbf{e}_t$ are available to the agent as additional inputs.

In addition to subtask reward defined in the task graph ($r_+$), the agent receives a time penalty for each step as a cost ($r_-$). To maximize the overall reward ($r = r_+ + r_-$), the agent needs to achieve the balance between two sources of rewards by minimizing costs while maximizing subtask rewards.

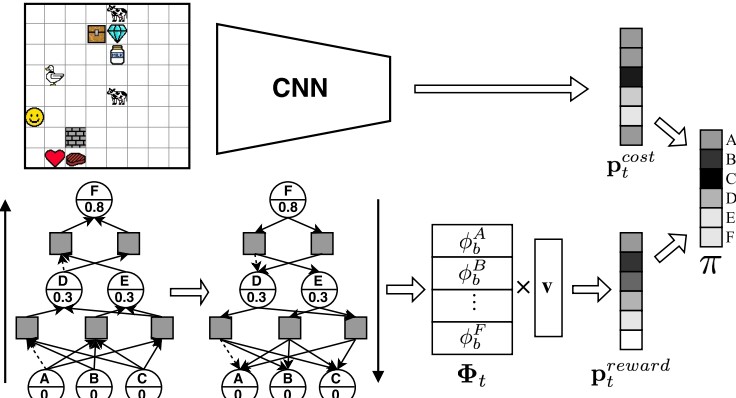

Figure 2: Neural task graph solver architecture. The task module encodes the task graph through a bottom-up and top-down process, and outputs the reward score ($\mathbf{p}_t^{reward}$). The observation module encodes observation using CNN and outputs the cost score ($\mathbf{p}_t^{cost}$). The final policy is a softmax policy over the sum of two scores.

Thus, the agent is required to take into account subtask dependencies in the task graph as well as observations to compute the cost of each subtask.

## 4 METHOD

We propose *neural task graph solver* (NTS) which is a neural network that encodes a task graph and an observation as shown in Figure 2. Our NTS is trained through actor-critic method to maximize the reward. To address the difficulty of training due to the complex nature of the problem, we propose *reward-propagation policy*, which propagates the reward information between related subtasks to model their dependencies. Since the reward-propagation policy acts as a reasonably good non-parametric policy, it is used to pre-train NTS through policy distillation. Section 4.1 describes the NTS architecture, and Section 4.2 describes how to construct the reward propagation policy.

### 4.1 NEURAL TASK GRAPH SOLVER

The NTS consists of two modules as illustrated in Figure 2: a *task module* and an *observation module*. The task module takes a task graph, a time budget, and a subtask completion indicator ($\mathbf{x}_t$) as input and produces a probability distribution over subtasks. Specifically, a recursive-reverse-recursive neural network (R3NN) (Parisotto et al. (2016)) is used to encode the task graph as follows:

$$\phi_{f,a}^i = f_{\theta_a}\left(\sum_{j \in Child_i} w_+^{i,j} \phi_{f,o}^j\right), \qquad \phi_{b,a}^i = b_{\theta_a}\left(\sum_{j \in Parent_i} \phi_{b,o}^j, \phi_{f,a}^i\right), \qquad (1)$$

$$\phi_{f,o}^i = f_{\theta_o}\left(\sum_{j \in Child_i} \phi_{f,a}^j, x_t^i, e^i, s\right), \qquad \phi_{b,o}^i = b_{\theta_o}\left(\sum_{j \in Parent_i} w_+^{i,j} \phi_{b,a}^j, \phi_{f,o}^i, r^i\right), \qquad (2)$$

where $w_+^{i,j} = -1$ if there is NOT connection between $j$-th OR node and $i$-th AND node and 1 otherwise, $\phi_{f,a}^i, \phi_{b,a}^i$ are the bottom-up and top-down embedding of $i$-th AND node respectively, and $\phi_{f,o}^i$ and $\phi_{b,o}^i$ are the bottom-up and top-down embedding of $i$-th OR node respectively. $Child_i, Parent_i$ represent a set of $i$-th node's children and parents respectively. $r^i \in \mathbb{R}$ is the reward when the $i$-th subtask is executed, and $s \in \mathbb{R}$ is the number of remaining steps, and $x_t^i \in \{0, 1\}$ is the subtask completion indicator. $f_\theta$ and $b_\theta$ are encoding functions for bottom-up and top-down recursive neural networks. Intuitively, a bottom-up recursive neural network is used to encode subtasks and preconditions, and a top-down recursive neural network is used to propagate information about future subtasks and rewards to children nodes. We use different parameters for AND/OR node encodings and multiply -1 to the embedding for NOT operation.

The embeddings are transformed to reward scores as follows:

$$\mathbf{p}_t^{reward} = \mathbf{\Phi}_t^\top \mathbf{v}, \qquad (3)$$

where $\boldsymbol{\Phi}_t = [\phi_b^1, \ldots, \phi_b^N] \in \mathbb{R}^{E \times N}$, and $\mathbf{v} \in \mathbb{R}^E$ is a weight vector for reward scoring. To sum up, the task module encodes the task graph using R3NN and estimates how good each subtask is.

The observation module encodes the input observation ($\mathbf{s}_t$) using a convolutional neural network (CNN) and outputs a cost score:

$$\mathbf{p}_t^{cost} = \text{CNN}(\mathbf{s}_t, s). \tag{4}$$

An ideal observation module would learn to estimate high scores for subtasks where the target object is close to the agent, because they would require less costs (i.e., time).

The NTS policy is a softmax policy which adds reward scores and cost scores computed from each module as follows:

$$\pi(\mathbf{o}_t|\mathbf{s}_t, \mathbf{G}, \mathbf{x}_t, s) = \text{Softmax}(\mathbf{p}_t^{reward} + \mathbf{p}_t^{cost}). \tag{5}$$

## 4.2 PRE-TRAINING NEURAL TASK GRAPH SOLVER FROM REWARD PROPAGATION POLICY

Let $\mathbf{r}_s \in \mathbb{R}^N$ be a vector of rewards of all subtasks. Let $\mathbf{x}_t$ be a subtask completion indicator vector and $\mathbf{e}_t$ be a eligibility vector at time-step $t$ (see Section 3 for definitions). Then, the sum of subtask reward until time-step $t$ is given as:

$$R_t = \mathbf{r}_s^T \mathbf{x}_t. \tag{6}$$

We first modify the reward formulation such that it gives a partial reward for satisfying preconditions to encourage the agent to satisfy precondition of a subtask with large reward. The sum of modified subtask reward is defined as:

$$\widetilde{R}_t = \mathbf{r}_s^T (\mathbf{x}_t + \mathbf{e}_t)/2. \tag{7}$$

Note that the agent receives a half of the subtask reward when it satisfies its precondition, and receives the rest of reward when it executes the subtask. The eligibility vector ($\mathbf{e}_t$) can be computed from the task graph and $\mathbf{x}_t$ as follows:

$$e_t^i = \underset{j \in Child_i}{\text{OR}} \left( y_{AND}^j \right), \tag{8}$$

$$y_{AND}^i = \underset{j \in Child_i}{\text{AND}} \left( \widehat{x}_t^{i,j} \right), \tag{9}$$

$$\widehat{x}_t^{i,j} = x_t^j w^{i,j} + (1 - x_t^j)(1 - w^{i,j}) \tag{10}$$

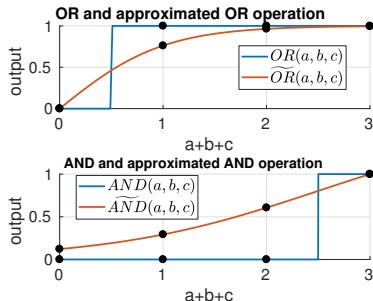

Figure 3: Visualization of OR, $\widetilde{\text{OR}}$, AND, and $\widetilde{\text{AND}}$ operations with three inputs (a,b,c).

where $y_{AND}^i$ is the output of $i$-th AND node, and $w^{i,j} = 0$ if there is a NOT connection between $i$-th node and $j$-th node, otherwise $w^{i,j} = 1$. Intuitively, $\widehat{x}_t^{i,j} = 1$ when $j$-th node does not violate the pre-condition of $i$-th node.

Note that $\tilde{R}_t$ is not differentiable with respect to $\mathbf{x}_t$ because AND($\cdot$) and OR($\cdot$) are not differentiable. To derive our reward-propagation policy, we propose to substitute AND($\cdot$) and OR($\cdot$) functions with "smoothed" functions $\widetilde{\text{AND}}$ and $\widetilde{\text{OR}}$ as follows:

$$\widetilde{e}_t^i = \underset{j \in Child_i}{\widetilde{\text{OR}}} \left( \widetilde{y}_{AND}^j \right), \qquad \widetilde{y}_{AND}^i = \underset{j \in Child_i}{\widetilde{\text{AND}}} \left( \widehat{x}_t^{i,j} \right), \tag{11}$$

where $\widetilde{\text{AND}}$ and $\widetilde{\text{OR}}$ were implemented as scaled sigmoid and tanh functions as illustrated by Figure 3 (see Appendix for details). With the smoothed operations, the sum of smoothed and modified reward is given as:

$$\widehat{R}_t = \mathbf{r}_s^T (\mathbf{x}_t + \widetilde{\mathbf{e}}_t)/2. \tag{12}$$

Finally, the reward-propagation policy is a softmax policy on the gradient of $\widehat{R}_t$ with respect to $\mathbf{x}_t$ as follows:

$$\pi(\mathbf{o}_t|\mathbf{G}, \mathbf{x}_t) = \text{Softmax}\left( \nabla_{\mathbf{x}_t} \widehat{R}_t \right) = \text{Softmax}\left( \frac{1}{2} \mathbf{r}_s^T \nabla_{\mathbf{x}_t} \widetilde{\mathbf{e}}_t \right). \tag{13}$$

Intuitively, the reward-propagation policy puts high probabilities over subtasks that are likely to increase the smoothed reward by a large margin at time $t$. Since this is a reasonably good policy that can be constructed on the fly without any learning, we propose to use the reward-propagation policy to pre-train our NTS through policy distillation.

## 5 EXPERIMENT

In the experiment, we investigated the following research questions:

- Does the reward-propagation policy outperform other heuristic baselines (e.g. greedy policy, etc)?
- Is the reward-propagation policy helpful for training NTS?
- Can NTS deal with complex task dependencies under delayed reward?
- Can NTS generalize well to unseen task graphs?
- How does NTS perform compared to MCTS?
- Can NTS be used to improve MCTS?

### 5.1 EXPERIMENTAL SETTING

**Environment** We developed an environment based on MazeBase (Sukhbaatar et al., 2015). An observation is represented as $64 \times 64$ RGB image. There are 10 types of objects: *Cow*, *Milk*, *Chicken*, *Egg*, *Diamond*, *Heart*, *Box*, *Meat*, *Block*, and *Ice*. The agent can take 6 primitive actions: *up*, *down*, *left*, *right*, *pickup*, *transform* and agent cannot moves on to the *block* cell. *Pickup* removes the object under the agent, and *transform* changes the object under the agent to *Ice*. The objects are randomly generated for each episode. The agent receives a time penalty (-0.1) for each step. The episode length (time budget) was randomly set for each episode in a range such that $60\% - 80\%$ of subtasks are executed on average for both training and testing.

**Subtask** The set of subtasks is $\mathcal{O} = \{pickup, transform\} \times \mathcal{X}$ where $\mathcal{X}$ corresponds to 8 types of objects above. As we discussed in Section 3, the agent chooses options which execute subtasks rather than primitive actions. We used a pre-trained subtask executer to implement subtask execution policy (see Appendix for details).

**Task Graph** The training set of task graphs consists of 4 layers of task dependencies. The testing set of task graphs consists of 4 or more layers of task dependencies with a larger number of subtasks. Task dependencies (AND, OR, and NOT) were randomly generated for each episode. In addition, we added the following components into task graphs to make the overall task more challenging:

- Distractor subtask: A subtask without any parent node in the task graph. Executing this kind of subtask may give an immediate reward but is sub-optimal in the long run.
- Negative distractor subtask: A subtask with only NOT connection to parent nodes in the task graph. Executing this subtask may give an immediate reward, but this would make other subtasks not executable.
- Delayed reward: The agent may receive little or zero reward for executing subtasks in the lower layers (i.e., subtasks with few or no pre-conditions). But, the agent should execute some of them to make other subtasks eligible.

More details of task graphs are described in the Appendix.

### 5.2 AGENTS

We evaluated the following policies:

- **Random**: A policy which executes any eligible subtask.
- **Greedy**: A policy which executes the eligible subtask with the largest reward.
- **Near-Optimal**: A near-optimal policy computed from exhaustive search on *eligible* subtasks.
- **RProp**: Our reward-propagation policy.
- **NTS-Scratch**: Our NTS trained with actor-critic from scratch.
- **NTS-RProp**: Our NTS distilled from reward-propagation policy and fine-tuned with actor-critic.

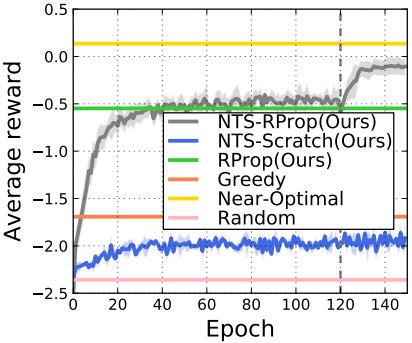

Figure 4: Learning curves. NTS-RProp is distilled from RProp until 120 epochs and fine-tuned through actor-critic after that.

| Task graph setting | | | | |
|---|---|---|---|---|
| Task | **D1** | **D2** | **D3** | **D4** |
| Depth | 4 | 4 | 5 | 6 |
| Number of subtask | 13 | 15 | 16 | 16 |
| Number of distractor | 3 | 4 | 3 | 0 |
| Performance ($R$) | | | | |
| NTS-RProp (Ours) | **.871** | **.701** | **.565** | **.380** |
| NTS-Scratch (Ours) | .131 | .084 | .108 | .139 |
| RProp (Ours) | .726 | .534 | .454 | .299 |
| Greedy | .267 | .194 | .205 | .216 |

Table 1: Generalization performance on unseen and larger task graphs. The task graphs in **D1** have the same graph structure as training set, but the graph was unseen. The task graphs in **D2**, **D3**, and **D4** have (unseen) larger graph structures. NTS-RProp outperforms other compared agents on all the task.

## 5.3 QUANTITATIVE RESULT

**Training Performance** The learning curve of each agent is shown in Figure 4. Our reward-propagation policy (RProp) significantly outperforms the greedy policy (Greedy) which executes the eligible subtask with the largest immediate reward. This implies that the proposed idea of back-propagating the reward gradient captures long-term dependencies among subtasks to some extent. The significant gap between NTS-RProp and NTS-Scratch in Figure 4 shows that the reward-propagation policy plays a key role in pre-training our NTS. We observed that NTS trained from scratch fails to capture complex task dependencies and only outperforms the random baseline. We believe that the reward-propagation policy gives a meaningful learning signal even if the reward is delayed by backpropagating the reward signal from the subtasks in the higher layers to the subtasks in the lower layers.

We also found that NTS-RProp further improves the performance through fine-tuning with actor-critic method. We hypothesize that our NTS learned to implicitly compute the expected costs of executing subtasks from the observations and consider them as well as task graphs.

**Generalization Performance** To investigate how different agents deal with unseen and larger task graphs, we measured performances on larger task graphs by varying the number of layers of the task graphs from 4 to 6 with a larger number of subtasks. Table 1 summarizes the results in terms of normalized reward $\bar{R} = (R - R_{min})/(R_{max} - R_{min})$ where $R_{min}$ and $R_{max}$ correspond to the average reward of the random and the near-optimal policy respectively. Though the performance degrades as the task graph becomes larger as expected, NTS-RProp generalizes well to larger task graphs and consistently outperforms all the other agents. This result indicates that the learned weights of `AND` and `OR` modules in NTS are general enough to capture more complex task dependencies in larger task graphs.

## 5.4 QUALITATIVE RESULT

Figure 5 visualizes an example of different agents' trajectories given the same initial observation and the task graph. As Greedy agent chooses the subtask that gives the largest reward among all eligible subtasks, it fails to execute the subtask 'F' at the highest layer within the time limit. In contrast, RProp agent receives a higher reward by executing the subtask 'F', which shows that it can consider the long-term effect of initial subtasks (e.g., 'A', 'B') on the later subtasks (e.g., 'D', 'E') through our reward-propagation method. Furthermore, our NTS-RProp agent found the optimal sequence of subtasks. Even though the optimal subtasks ('B-C-E-F') give a smaller amount of rewards compared to RProp agent's trajectory in the task graph, they require much less costs (i.e., time) to execute. This demonstrates that our NTS considers not only the task graph but also the expected costs for executing each subtask from the observation to make a better decision. Figure 6 visualizes more complicated example of trajectories.

## 5.5 ANALYSIS OF TASK GRAPH COMPONENTS

To investigate how agents deal with different types of task graph components, we evaluated all agents on the following types of task graphs:

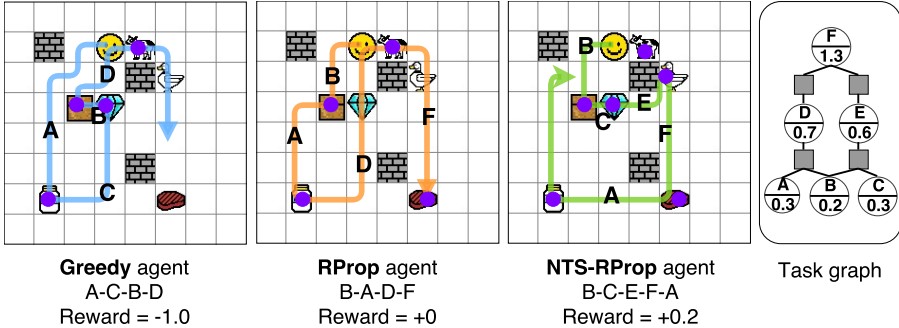

Figure 5: Example trajectories of Greedy, RProp, and NTS-RProp agents given 25 steps. Greedy agent fails to execute the subtask 'F' which gives the largest reward within the time limit, whereas RProp and NTS-RProp agents execute them by executing its pre-conditions. NTS-RProp agent found a shorter trajectory of subtasks, and executed more subtasks within the time limit than the other agents (e.g., 5 compared to 4).

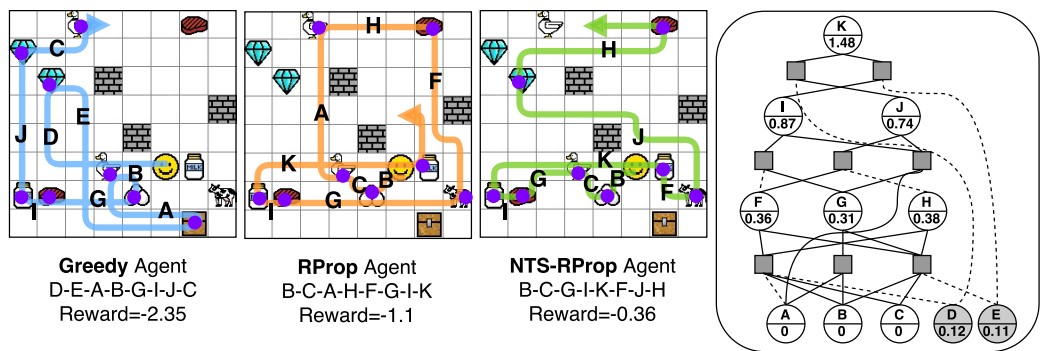

Figure 6: More complicated example trajectories of Greedy, RProp, and NTS-RProp agents given 45 steps. The task graph includes NOT operation and Neg-Distractor (subtask D and E). Greedy agent executes the negative-distractors since they give positive immediate rewards, which makes it impossible to execute the subtask 'K' which gives the largest reward. RProp and NTS-RProp agents avoid negative-distractors and successfully execute subtask 'K' by satisfying its pre-conditions. NTS-RProp agent found a shorter path to execute subtask 'K' in the task graph, while RProp found a sub-optimal path to execute subtask 'K'.

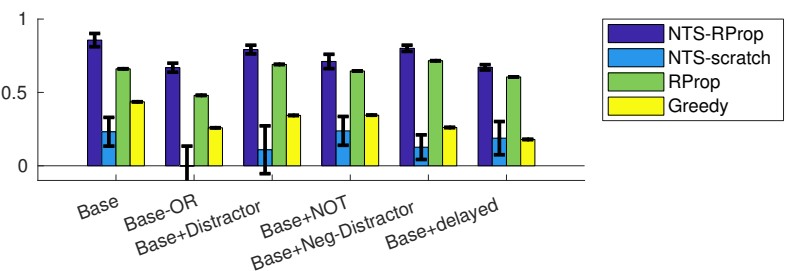

Figure 7: Normalized performance on task graphs with different types of dependencies.

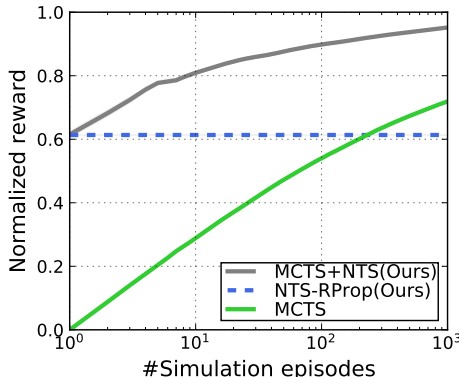

Figure 8: Performance of MCTS+NTS and MCTS on **D2** (see Table 1) per the number of simulated episodes. NTS-RProp performs as well as MCTS with 231 simulated episodes. MCTS augmented with NTS significantly outperforms MCTS.

- 'Base' set consists of task graphs with only `AND` and `OR` operation.
- 'Base-OR' set removes all the `OR` operations from the base set.
- 'Base+Distractor' set adds several distractor subtasks to the base set.
- 'Base+NOT' set adds several `NOT` operations to the base set.
- 'Base+NegDistractor' set adds several negative distractor subtasks to the base set.
- 'Base+Delayed' set assigns zero reward to all subtasks but the top-layer subtask.

The results are shown in Figure 7. Since 'Base' and 'Base-OR' sets do not contain `NOT` operation and every subtask gives a positive reward, the greedy baseline performs reasonably well compared to other sets of task graphs. It is also shown that the gap between NTS-RProp and RProp is relatively large in these two sets. This is because computing the optimal ordering between subtasks is more important in these kinds of task graphs. Since only NTS-RProp can take into account the cost of each subtask from the observation, it can find a better sequence of subtasks more often.

In 'Base+Distractor', 'Base+NOT', and 'Base+NegDistractor' cases, it is more important for the agent to carefully find and execute subtasks that have a positive effect in the long run while avoiding distractors that are not helpful for executing future subtasks. In these tasks, the greedy baseline tends to execute distractors very often because it cannot consider the long-term effect of each subtask in principle. On the other hand, our RProp can naturally screen out distractors by getting zero or negative gradient during reward back-propagation. Similarly, RProp performs well on 'Base+Delayed' set because it gets non-zero gradients for all subtasks that are connected to the final rewarding subtask. Since our NTS-RProp was distilled from RProp, it can handle delayed reward or distractors as well as (or better than) RProp.

## 5.6 COMBINING NTS WITH MONTE-CARLO TREE SEARCH

In this section, we investigated how well our NTS agent performs compared to conventional search-based methods and how our NTS agent can be combined with search-based methods to further improve the performance. More specifically, we implemented and evaluated the following methods:

- MCTS: An MCTS algorithm with UCB (Auer et al., 2002) criterion for choosing an action at each node in the search tree (see Appendix for the detail).
- MCTS+NTS: An MCTS algorithm combined with our NTS-RProp agent. NTS-RProp policy was used as a rollout policy to explore reasonably good states during tree search, which is similar to AlphaGo (Silver et al., 2016) (see Appendix for the detail).

The results are shown in Figure 8. It turns out that our NTS-RProp performs as well as MCTS method with approximately 231 simulated episodes on average. This indicates that NTS-RProp

can efficiently find reasonably good solutions without requiring any lookahead search. More interestingly, NTS combined with MCTS (i.e., MCTS+NTS) significantly outperforms MCTS and achieves approximately $0.95$ normalized reward with $1,000$ simulated episodes. We found that the Near-Optimal policy, which corresponds to normalized reward of $1.0$, uses approximately $600,000$ simulated episodes. Thus, MCTS augmented by NTS performs almost as well as the exhaustive search by searching over only $0.16\%$ sequences of subtasks among all possible sequences of subtasks. This result implies that NTS can also be used to improve conventional search-based methods by effectively reducing the search space.

## 6 CONCLUSION

We introduced the task graph execution problem which is an effective and principled way of describing many complex tasks. To address the difficulty of dealing with complex task dependencies, we proposed a reward-propagation policy derived from a differentiable form of task graph, which plays an important role in pre-training our neural task graph solver architecture. The empirical results showed that our agent can deal with long-term dependencies between subtasks and generalize well to unseen task graphs. In addition, we showed that our agent can be used to effectively reduce the search space of MCTS so that the agent can find a near-optimal solution with a small number of simulations.

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

## A  DETAILS OF NTS ARCHITECTURE

**Task module**  was implemented with four submodules: forward-OR-network (fORnet), forward-AND-network (fANDnet), backward-OR-network (bORnet), and backward-AND-network (bAND-net). The input and output of each submodule is defined by Eq. 1 and 2.

$$\phi_{f,a}^i = f_{\theta_a} \left( \sum_{j \in Child_i} w_+^{i,j} \phi_{f,o}^j \right), \qquad \phi_{b,a}^i = b_{\theta_a} \left( \sum_{j \in Parent_i} \phi_{b,o}^j, \phi_{f,a}^i \right), \qquad (14)$$

$$\phi_{f,o}^i = f_{\theta_o} \left( \sum_{j \in Child_i} \phi_{f,a}^j, x_t^i, e^i, s \right), \qquad \phi_{b,o}^i = b_{\theta_o} \left( \sum_{j \in Parent_i} w_+^{i,j} \phi_{b,a}^j, \phi_{f,o}^i, r^i \right), \qquad (15)$$

Each submodule takes the output embeddings from its children nodes, and take element-wise sum over all input embeddings giving single 128-dimensional vector, while fANDnet and bORnet multiplies $w_+^{i,j}$ before summation to deal with NOT operation in the task graph. Then, the summed embedding is concatenated with all additional input as defined in Eq. 14 and 15, which is further transformed with three fully-connected layers with 128 units. The last fully-connected layer outputs 128-dimensional output embedding. Then, $\phi_{b,o}^i$'s are transformed to reward scores as Eq. 3 with FC(1) layer and reward baseline scores with another FC(1) layer. We used parametric ReLU (PReLU) function as activation function.

**Observation module**  The network consists of BN1-Conv1(16x8x8-8/0)-BN2-Conv2(32x3x3-1/1)-BN3-Conv3(32x3x3-1/1)-BN4-Conv4(32x3x3-1/1)-BN5-Conv5(32x3x3-1/1)-concat-FC(256)-FC(8), where the output embedding of Conv5 was concatenated with number of remaining time step $s$. Another network with exactly the same architecture except the last layer with FC(1) layer was used to output cost baseline scores. The output of FC(8) corresponds to the cost score of each object. Each score value is assigned to the corresponding subtask before it is summed with reward score. Finally the policy of NTS is calculated by adding reward score and cost score, and taking softmax. The baseline output is obtained by adding reward baseline score and cost baseline score.

## B  DETAILS OF LEARNING NTS AGENT

**Learning objectives**  The NTS architecture is first trained through policy distillation and fine-tuned using actor-critic method with generalized advantage estimation.

During policy distillation, the KL divergence between NTS and teacher policy (RProp) is minimized as follows:

$$\nabla_\theta \mathcal{L}_1 = \mathbb{E}_{G \sim \mathcal{D}} \left[ \mathbb{E}_{s \sim \pi_\theta^G} \left[ \nabla_\theta D_{KL} \left( \pi_T^G || \pi_\theta^G \right) \right] \right], \qquad (16)$$

where $\theta$ is the parameter of NTS architecture, $\pi_\theta^G$ is the simplified notation of NTS policy with task graph input $G$, $\pi_T^G$ is the simplified notation of teacher (RProp) policy with task graph input $G$, $D_{KL} \left( \pi_T^G || \pi_\theta^G \right) = \sum_a \pi_T^G(a|s) \log \frac{\pi_T^G(a|s)}{\pi_\theta^G(a|s)}$ and $\mathcal{D} \subset \mathcal{G}$ is the training set of task graphs. For both policy distillation and fine-tuning, we sampled one task graph for each 16 parallel workers, and each worker in turn sample a mini-batch of 16 world configurations (maps). Then, NTS generates total 256 episodes in parallel. After generating episode, the gradient from 256 episodes are collected and averaged, and then back-propagated to update the parameter. For policy distillation, we trained NTS for 120 epochs where each epoch involves 100 times of update. After policy distillation, we fine-tune NTS agent using actor-critic method with generalized advantage estimation (GAE) (Schulman et al., 2015) as follows:

$$\nabla_\theta \mathcal{L}_2 = \mathbb{E}_{G \sim \mathcal{D}} \left[ \mathbb{E}_{s \sim \pi_\theta^G} \left[ -\nabla_\theta \log \pi_\theta \left( \mathbf{o}_t | \mathbf{s}_t, \mathbf{G}, \mathbf{x}_t, s \right) \sum_{k=0}^{\infty} (\gamma \lambda)^k \delta_{t+l} \right] \right], \qquad (17)$$

where $\gamma \in [0, 1]$ is a discount factor, $\lambda \in [0, 1]$ is a weight for balancing between bias and variance of the advantage estimation, and $\delta_t = r_t + \gamma V^\pi(s_{t+1}; \theta') - V^\pi(\mathbf{s}_t; \theta')$. During training, we update $\theta'$ to minimize $\mathbb{E} \left[ (R_t - V^\pi(s_t; \theta'))^2 \right]$.

**Hyperparameters** For both finetuning and policy distillation, we used RMSProp optimizer with the smoothing parameter of 0.97 and epsilon of 1e-6. When distilling agent with teacher policy, we used learning rate=5e-5 and 2.5e-8 for finetuning, respectively. For training NTS-scratch, we used learning rate=2.5e-6. For actor-critic training for both NTS-scratch and NTS-Rprop, we used $\alpha = 0.01$. For better exploration, we applied entropy regularization with a weight of 0.1 for NTS-scratch and 0.05 for finetuning NTS-RProp and linearly decreased it to zero for the first 10,000 iterations for NTS-scratch and 1,000 for finetuning NTS-RProp. The total number of iterations was 15,000 for both NTS-scratch. The total number of iterations for ditilling NTS-RProp was 12,000 and 3,000 for finetuning.

## C  DETAILS OF AND/OR OPERATION AND APPROXIMATED AND/OR OPERATION

In Eqs. 8 and 9, the output of $i$-th AND and OR node in task graph were defined represented using AND and OR operation with multiple input. They can be represented in logical expression as below:

$$\underset{j \in Child_i}{\text{OR}} \left( y^j \right) = y^{j_1} \vee y^{j_2} \vee \ldots \vee y^{j_{|Child_i|}}, \tag{18}$$

$$\underset{j \in Child_i}{\text{AND}} \left( y^j \right) = y^{j_1} \wedge y^{j_2} \wedge \ldots \wedge y^{j_{|Child_i|}}, \tag{19}$$

where $j_1, \ldots, j_{|Child_i|}$ are the elements of a set $Child_i$ and $Child_i$ is the set of inputs coming from the children nodes of $i$-th node. Then, these AND and OR operations are smoothed as below:

$$\underset{j \in Child_i}{\widetilde{\text{OR}}} \left( \widetilde{y}^j_{AND} \right) = h_{or} \left( \sum_{j \in Child_i} \widetilde{y}^j_{AND} \right), \tag{20}$$

$$\underset{j \in Child_i}{\widetilde{\text{AND}}} \left( \widehat{x}^{i,j}_t \right) = h_{and} \left( \sum_{j \in Child_i} \widehat{x}^{i,j}_t - |Child_i| + 0.5 \right), \tag{21}$$

where $h_{or}(x) = \alpha_o \tanh(x/\beta_o)$, $h_{and}(x) = \alpha_a \sigma(x/\beta_a)$, $\sigma(\cdot)$ is sigmoid function, and $\alpha_o, \beta_o, \alpha_a, \beta_a \in \mathbb{R}$ are hyperparameters to be set. We used $\beta_a = 2, \beta_o = 1, \alpha_a = 1/\sigma(0.25)$, and $\alpha_o = 1$

## D  DETAILS OF SUBTASK EXECUTOR

**Architecture** The subtask executor has the same architecture of the parameterized skill architecture of Oh et al. (2017) with slightly different hyperparameters. The network consists of Conv1(16x8x8-8/0)-Conv2(32x1x1-1/0)-Conv3(32x3x3-1/1)-Conv4(32x3x3-1/1)–Conv5(32x3x3-1/1)-LSTM(256)-FC(256). The subtask executor takes two task parameters ($q = [q^{(1)}, q^{(2)}]$) as additional input and computes $\chi(q) = ReLU(W^{(1)}q^{(1)} \odot W^{(2)}q^{(2)})$ to compute the subtask embedding, and further linearly transformed into the weights of Conv3 and the (factorized) weight of LSTM through multiplicative interaction as described above. Finally, the network has three fully-connected output layers for actions, termination probability, and baseline, respectively.

**Learning objective** The subtask executor is trained through policy distillation and then finetuned. Similar to (Oh et al., 2017), we first trained 16 teacher policy network for each subtask. The teacher policy network consists of Conv1(8x8x8-8/0)-BN1(8)-Conv2(16x3x3-1/1)-BN2(16)-Conv3(16x3x3-1/1)-BN3(16)-Conv4(16x3x3-1/1)-BN4(16)-LSTM(128)-FC(128). Similar to subtask executor network, the teacher policy network has three fully-connected output layers for actions, termination probability, and baseline, respectively. Then, the learned teacher policy networks are used as teacher policy for policy distillation to train subtask executor. During policy distillation, we train agent to minimize the following objective function:

$$\nabla_\xi \mathcal{L}_{1,sub} = \mathbb{E}_{\mathbf{o} \sim \mathcal{O}} \left[ \mathbb{E}_{s \sim \pi^o_\xi} \left[ \nabla_\xi \left\{ D_{KL} \left( \pi^o_T || \pi^o_\xi \right) \right\} + \alpha L_{term} \right] \right], \tag{22}$$

where $\xi$ is the parameter of subtask executor network, $\pi^o_\xi$ is the simplified notation of subtask executor given input subtask $\mathbf{o}$, $\pi^o_T$ is the simplified notation of teacher policy for subtask $\mathbf{o}$,

$L_{term} = -\mathbb{E}_{s_t \in \tau_{\mathbf{o}}} [\log \beta_\xi(s_t, \mathbf{o})]$ is the cross entropy loss of predicting termination, $\tau_{\mathbf{o}}$ is a set of state in which the subtask $\mathbf{o}$ is terminated, $\beta_\xi(s_t, \mathbf{o})$ is the termination probability output, and $D_{KL}\left(\pi_T^{\mathbf{o}}||\pi_\xi^{\mathbf{o}}\right) = \sum_a \pi_T^{\mathbf{o}}(a|s) \log \frac{\pi_T^{\mathbf{o}}(a|s)}{\pi_\xi^{\mathbf{o}}(a|s)}$. After policy distillation, we fine-tuned subtask executor using actor-critic method with generalized advantage estimation (GAE):

$$\nabla_\xi \mathcal{L}_{2,sub} = \mathbb{E}_{\mathbf{o}\sim\mathcal{O}} \left[ \mathbb{E}_{s\sim\pi_\xi^{\mathbf{o}}} \left[ -\nabla_\xi \log \pi_\xi \left(\mathbf{a}_t|\mathbf{s}_t, \mathbf{o}\right) \sum_{k=0}^{\infty} (\gamma\lambda)^k \delta_{t+l} + \alpha\nabla_\xi L_{term} \right] \right], \qquad (23)$$

where $\gamma \in [0,1]$ is a discount factor, $\lambda \in [0,1]$ is a weight for balancing between bias and variance of the advantage estimation, and $\delta_t = r_t + \gamma V^\pi(s_{t+1}; \xi') - V^\pi(\mathbf{s}_t; \xi')$. We used $\alpha = 0.1$ for both policy distillation and fine-tuning.

## E    DETAILS OF SEARCH ALGORITHMS

Each iteration of Monte-Carlo tree search method consists of four stages: selection-expansion-rollout-backpropagation. For selection, we used UCB criterion [Auer et al. (2002)]. Specifically, the option for which the score below has the highest value is chosen for selection:

$$\text{score} = \frac{R_i}{n_i} + C\sqrt{\frac{\ln N}{n_i}}, \qquad (24)$$

where $R_i$ is the accumulated return at $i$-th node, $n_i$ is the number of visit of $i$-th node, $C$ is the exploration-exploitation balancing weight, and $N$ is the number of total iterations so far. We found that $C = 2\sqrt{2}$ gives the best result and used it for both MCTS and MCTS+NTS methods. For expansion, MCTS randomly chooses the remaining eligible subtask, while the subtask is chosen by NTS-RProp policy for MCTS+NTS method. More specifically, the subtask for which the NTS-RProp policy probability has the highest value is chosen for MCTS+NTS method. Due to a memory limit, the expansion of search tree was truncated at the depth of 7 and performed rollout after the maximum depth. In rollout, MCTS randomly executes an eligible subtask, while MCTS+NTS execute again the subtask with the highest NTS-RProp policy probability. Once the episode is terminated, the result is back-propagated; the estimated value function and visit count are updated for each node in the tree.

## F    DETAILS OF TASK GENERATION

| | | |
|---|---|---|
| | $N_T$ | number of tasks in each layer |
| Nodes | $N_D$ | number of distractors in each layer |
| | $N_A$ | number of AND node in each layer |
| | $N_{ac}^+$ | number of children of AND node in each layer |
| | $N_{ac}^-$ | number of children of AND node with NOT connection in each layer |
| Edges | $N_{dp}$ | number of parents with NOT connection of distractors in each layer |
| | $N_{oc}$ | number of children of OR node in each layer |
| Episode | $N_{step}$ | number of step given for each episode |

Table 2: Parameters for generating task including task graph parameter and episode length.

For training and test sample generation, the reward graph structure was defined in terms of the parameters in table 2. To cover wide range of task graphs, we randomly sampled the parameters $N_A, N_O, N_{ac}^+, N_{ac}^-, N_{dc}$, and $N_{oc}$ in a reasonable range, while $N_T$ and $N_D$ was manually set. We only generated a graph without duplicated AND nodes with same children node. We carefully set the range of each parameter such that at least 1,000 different task graph can be generated with given parameters. The table 3 and 4 summarizes parameters used to generate training and evaluation task graphs.

| | | |
|---|---|---|
| Train (=**D1**) | $N_T$ | {6,4,2,1} |
| | $N_D$ | {2,1,0,0} |
| | $N_A$ | {3,3,2}-{4,3,3} |
| | $N_{ac}^+$ | {1,1,2}-{3,2,2} |
| | $N_{ac}^-$ | {0,0,0}-{3,3,1} |
| | $N_{dp}$ | {0,0,0}-{2,2,0} |
| | $N_{oc}$ | {1,1,1}-{2,2,2} |
| | $N_{step}$ | 40-60 |
| **D2** | $N_T$ | {7,5,2,1} |
| | $N_D$ | {2,2,0,0} |
| | $N_A$ | {4,3,2}-{5,3,3} |
| | $N_{ac}^+$ | {1,1,2}-{3,2,2} |
| | $N_{ac}^-$ | {0,0,0}-{3,3,1} |
| | $N_{dp}$ | {0,0,0,0}-{3,3,0,0} |
| | $N_{oc}$ | {1,1,1}-{2,2,2} |
| | $N_{step}$ | 40-60 |
| **D3** | $N_T$ | {5,4,4,2,1} |
| | $N_D$ | {1,1,1,0,0} |
| | $N_A$ | {3,3,2,2}-{4,3,3,3} |
| | $N_{ac}^+$ | {1,1,1,2}-{3,2,2,2} |
| | $N_{ac}^-$ | {0,0,0,0}-{3,1,1,0} |
| | $N_{dp}$ | {0,0,0,0,0}-{3,3,3,0,0} |
| | $N_{oc}$ | {1,1,1,1}-{2,2,2,2} |
| | $N_{step}$ | 45-65 |
| **D4** | $N_T$ | {4,3,3,3,2,1} |
| | $N_D$ | {0,0,0,0,0,0} |
| | $N_A$ | {4,3,3,2,2}-{4,3,4,3,3} |
| | $N_{ac}^+$ | {1,1,1,1,2}-{3,2,3,2,2} |
| | $N_{ac}^-$ | {0,0,0,0,0}-{3,1,1,0,0} |
| | $N_{dp}$ | {0,0,0,0,0,0}-{0,0,0,0,0,0} |
| | $N_{oc}$ | {1,1,1,1,1}-{2,2,2,2,2} |
| | $N_{step}$ | 45-65 |

Table 3: Task graph parameters for training set and tasks **D1**∼**D4**.

| | | |
|---|---|---|
| **Base** | $N_T$ | {4,3,2,1} |
| | $N_D$ | {0,0,0,0} |
| | $N_A$ | {3,3,2}-{4,3,3} |
| | $N_{ac}^+$ | {1,1,2}-{3,2,2} |
| | $N_{ac}^-$ | {0,0,0}-{0,0,0} |
| | $N_{dp}$ | {0,0,0,0}-{0,0,0,0} |
| | $N_{oc}$ | {1,1,1}-{2,2,2} |
| | $N_{step}$ | 40-60 |
| **-OR** | $N_{oc}$ | {1,1,1}-{1,1,1} |
| **+Distractor** | $N_D$ | {2,1,0,0} |
| **+NOT** | $N_{ac}^+$ | {0,0,0}-{3,2,2} |
| **+NegDistractor** | $N_D$ | {2,1,0,0} |
| | $N_{dp}$ | {0,0,0,0}-{3,3,0,0} |
| **+Delayed** | $r_s$ | {0,0,0,1.6}-{0,0,0,1.8} |

Table 4: Task graph parameters for analysis of task graph components.

