# OpenReview forum: "Neural Task Graph Execution"
_ICLR.cc/2018/Conference — Reject_

### Official Review · AnonReviewer1 · 2017-11-27
**Neural Task Graph Execution**

**Rating:** 6
**Confidence:** 4

**Review:**

In the context of multitask reinforcement learning, this paper considers the problem of learning behaviours when given specifications of subtasks and the relationship between them, in the form of a task graph. The paper presents a neural task graph solver (NTS), which encodes this as a recursive-reverse-recursive neural network. A method for learning this is presented, and fine tuned with an actor-critic method. The approach is evaluated in a multitask grid world domain.

This paper addresses an important issue in scaling up reinforcement learning to large domains with complex interdependencies in subtasks. The method is novel, and the paper is generally well written. I unfortunately have several issues with the paper in its current form, most importantly around the experimental comparisons.

The paper is severely weakened by not comparing experimentally to other learning (hierarchical) schemes, such as options or HAMs. None of the comparisons in the paper feature any learning. Ideally, one should see the effect of learning with options (and not primitive actions) to fairly compare against the proposed framework. At some level, I question whether the proposed framework is doing any more than just value function propagation at a task level, and these experiments would help resolve this.

Additionally, the example domain makes no sense. Rather use something more standard, with well-known baselines, such as the taxi domain.

I would have liked to see a discussion in the related work comparing the proposed approach to the long history of reasoning with subtasks from the classical planning literature, notably HTNs.

I found the description of the training of the method to be rather superficial, and I don't think it could be replicated from the paper in its current level of detail.

The approach raises the natural questions of where the tasks and the task graphs come from. Some acknowledgement and discussion of this would be useful.

The legend in the middle of Fig 4 obscures the plot (admittedly not substantially).

There are also a number of grammatical errors in the paper, including the following non-exhaustive list:
2: as well as how to do -> as well as how to do it
Fig 2 caption: through bottom-up -> through a bottom-up
3: Let S be a set of state -> Let S be a set of states
3: form of task graph -> form of a task graph
3: In addtion -> In addition
4: which is propagates -> which propagates
5: investigated following -> investigated the following

---

> ### Author Response · Authors · 2017-12-18
> **Author response**
>
> Thank you for the constructive comment.
> We’ve posted a common response to the all reviewers as a separate comment above.
> We’d appreciate it if you go through the common response as well as this comment.
>
> Q1) Learning with/without options for fair comparison
> A) We assume that a pre-trained subtask executor (that can perform a variety of tasks) is available for all methods. Here, we can view each instantiation of the subtask as an option, and we consider learning an optimal policy to execute task graphs using such options. Since we used options for all methods including all baselines, we believe that this is a fair comparison.
>
>
>
> Q2) The proposed framework is doing any more than just value function propagation at a task level.
> A) We are not clear about what you mean by “value function propagation at a task level”. Would you please give us the specific reference on prior work and clarify this comment in more details? Intuitively speaking, our “reward-propagation policy” is indeed designed to propagate reward function in the task graph, and we believe showing how it can be done with a concrete algorithm is one of our contributions. Furthermore, our final NTS architecture improves over the “reward-propagation” method by combining all relevant information (e.g., observations, task graph embedding, and prior experience) together.
>
> If the question is whether the learned policy is trivial, we demonstrate both qualitative and quantitative results showing that the learned strategy considers long-term dependencies between many subtasks. To our knowledge, this is infeasible for most traditional RL methods.
> We would also appreciate if you take a look at the new results (Section 5.6) from the current revision. Specifically, we show how well our NTS performs compared to a sophisticated search/planning method (MCTS). It turns out that NTS (without any search) performs as well as MCTS with approximately 250 simulated search episodes. Combining NTS with MCTS, we further improve the performance. These results suggest that the learned policy of NTS is very strong and efficient.
>
>
>
> Q3) Related work on classical planning
> A) Thank you for pointing out a relevant work. We discussed HTN in the revision. In brief, HTN considers a similar planning problem in that a planner should find the optimal sequence of tasks to minimize the overall cost. The main differences are the following:
> 1) Our problem does not have a particular goal task but is an episodic RL task with a finite horizon, so the agent should consider all possible sequences of tasks to maximize the reward within a time limit rather than computing an optimal path toward a particular goal task.
> 2) HTN assumes that a task graph describes all necessary information for planning, whereas our task graph does not have cost information, and the agent should implicitly “infer” cost information from the observation. The observation module of our NTS plays a key role for this.
> Due to these differences, HTN is not directly applicable to our problem.
>
>
>
> Q4) Domain
> A) Regarding your comments on our example domain, we agree that it could have been made more interesting and practically relevant. At the same time, we believe that our framework is general enough to be applicable to more interesting scenarios (e.g., cooking, cleaning, assemblies) with small changes in semantics, which we plan as future work. Regarding your comments on the taxi domain, our domain is a richer superset of the taxi domain. However, our typical experimental setup is much more challenging, and traditional hierarchical RL baselines are not applicable due to changing task graphs during testing. For details, please see our common response to all reviewers.
>
>
> Q5) Source of task graphs
> A) In our paper, we assumed that the task graph is given. However, in the future work, we plan to extend to scenarios when the reward is unknown or when the task graph structure is unknown. Note that these settings are extremely challenging for complex task dependencies, but we hypothesize that such unknowns (e.g., rewards and/or graph structures) might be also learnable through experience. For example, in case of the household robot example in the introduction of our paper, they may be learned through interaction with a user in a trial-and-error manner. These are well beyond the scope of the current work.

---

### Official Review · AnonReviewer3 · 2017-11-28
**Nice idea**

**Rating:** 6
**Confidence:** 3

**Review:**


Summary: the paper proposes an idea for multi-task learning where tasks have shared dependencies between subtasks as task graph. The proposed framework, task graph solver (NTS), consists of many approximation steps and representations: CNN to capture environment states, task graph parameterization, logical operator approximation; the idea of reward-propagation policy helps pre-training. The framework is evaluated on a relevant multi-task problem.

In general, the paper proposes an idea to tackle an interesting problem. It is well written, the idea is well articulated and presented. The idea to represent task graphs are quite interesting. However it looks like the task graph itself is still simple and has limited representation power. Specifically, it poses just little constraints and presents no stochasticity (options result in stochastic outcomes).

The method is evaluated in one experiment with many different settings. The task itself is not too complex which involves 10 objects, and a small set of deterministic options. It might be only complex when the number of dependency layer is large. However, it's still more convinced if the paper method is demonstrated in more domains.


About the description of problem statement in Section 3:

- How the MDP M and options are defined, e.g. transition functions, are tochastic?

- What is the objective of the problem in section 3

Related work: many related work in robotics community on the topic of task and motion planning (checkout papers in RSS, ICRA, IJRR, etc.) should also be discussed.

---

> ### Author Response · Authors · 2017-12-18
> **Author response**
>
> Thank you for the constructive comment.
> We’ve posted a common response to the all reviewers as a separate comment above.
> We’d appreciate it if you go through the common response as well as this comment.
>
>
> Q1) Task graph has limited representation power.
> A) We would like to point out that our task graph can represent any logical expression as it follows sum-of-product (SoP) form, which is widely used for logic circuit design. In addition, the task graph can be very expressive by forming a deep hierarchical structure of task dependencies. In other words, a precondition can be “deep” such as AND(A, B, NOT(OR(C, AND(D, E, OR(...))))). This provides a richer form of task descriptions and subsumes many existing tasks (e.g., Taxi domain, sequential instructions in [1])
>
> [1] Oh, et.al. (2017). Zero-shot task generalization with multi-task deep reinforcement learning.
>
> Q2) Stochasticity and multiple domains
> A) We agree with the reviewer that it would be interesting to introduce stochasticity in the environment (e.g., stochastic options), and showing results on multiple domains would make the paper stronger. We are working on this extension.
>
> We believe that the main contribution of this work is 1) to propose a richer and general form of task descriptions (task graph) compared to the previous work on multi-task RL and 2) to propose a deep RL architecture and reward-propagation policy for learning to find optimal solutions of any arbitrary task graphs and observations.
>
>
> Q3) How the MDP M and options are defined, e.g. transition functions, are stochastic?
> A) In our problem formulation, transition functions and reward functions of MDP can be either deterministic or stochastic. In our experiment, we focused on the case where both the transition and reward function are deterministic. Options used in the experiment are O = {pickup, transform} × X where X corresponds to 8 types of objects.
>
>
>
> Q4) What is the objective of the problem in section 3?
> A) The goal is to learn a multi-task policy \pi: S x G -> O that maximizes the overall reward (r=r_{+} + r_{-}), where S is a set of observations (input pixel image, remaining number of step, subtask completion indicator x_t, and eligibility vector e_t) and G is a set of task graphs, and O is a set of options available to the agent. We clarified this in the paper.
>
>
>
> Q5) Related work on motion planning in robotics
> A) Thank you for pointing out the relevant work. We added more papers on motion planning in the related work section. Please let us know if there is missing relevant work.

---

### Official Review · AnonReviewer4 · 2017-12-15

**Rating:** 4
**Confidence:** 4

**Review:**

This paper proposes to train recursive neural network on subtask graphs in order to execute a series of tasks in the right order, as is described by the subtask graph's dependencies. Each subtask execution is represented by a (non-learned) option. Reward shaping allows the proposed model to outperform simpler baselines, and experiments show the model generalizes to unseen graphs.

While this paper is as far as I can tell novel in how it does what it does, the authors have failed to convey to me why this direction of research is relevant.
- We know finding options is the hard part about options
- We already have good algorithms that take subtask graphs and execute them in the right order from the planning litterature

An interesting avenue would be if the subtask graphs were instead containing some level of uncertainty, or representing stochasticity, or anything that more traditional methods are unable to deal with efficiently, then I would see a justification for the use of neural networks. Alternatively, if the subtask graphs were learned instead of given, that would open the door to scaling an general learning. Yet, this is not discussed in the paper.

Another interesting avenue would be to learn the options associated with each task, possibly using the information from the recursive neural networks to help learn these options.


The proposed algorithm relies on fairly involved reward shaping, in that it is a very strong signal of supervision on what the next action should be. Additionaly, it's not clear why learning seems to completely "fail" without the pre-trained policy. The justification given is that it is "to address the difficulty of training due to the complex nature of the problem" but this is not really satisfying as the problems are not that hard. This also makes me question the generality of the approach since the pre-trained policy is rather simple while still providing an apparently strong score.


In your experiments, you do not compare with any state-of-the-art RL or hierarchical RL algorithm on your domain, and use a new domain which has no previous point of reference. It it thus hard to properly evaluate your method against other proposed methods.

What the authors propose is a simple idea, everything is very clearly explained, the experiments are somewhat lacking but at least show an improvement over more a naive approach, however, due to its simplicity, I do not think that this paper is relevant for the ICLR conference.

Comments:
- It is weird to use both a discount factor \gamma *and* a per-step penalty. While not disallowed by theory, doing both is redundant because they enforce the same mechanism.
- It seems weird that the smoothed logical AND/OR functions do not depend on the number of inputs; that is unless there are always 3 inputs (but it is not explained why; logical functions are usually formalised as functions of 2 inputs) as suggested by Fig 3.
- It does not seem clear how the whole training is actually performed (beyond the pre-training policy). The part about the actor-critic learning seems to lack many elements (whole architecture training? why is the policy a sum of "p^{cost}" and "p^{reward}"? is there a replay memory? How are the samples gathered?). (On the positive side, the appendix provides some interesting details on the tasks generations to understand the experiments.)
- The experiments cover different settings with different task difficulties. However, only one type of tasks is used. It would be good to motivate (in addition to the paragraph in the intro) the cases where using the algorithm described in the paper may be (or not?) the only viable option and/or compare it to other algorithms. Even tough not mandatory, it would also be a clear good addition to also demonstrate more convincing experiments in a different setting.
- "The episode length (time budget) was randomly set for each episode in a range such that 60% − 80% of subtasks are executed on average for both training and testing." --> this does not seem very precise: under what policy is the 60-80% defined? Is the time budget different for each new generated environment?
- why wait until exactly 120 epochs for NTS-RProp before fine-tuning with actor-critic? It seems that much less would be sufficient from figure 4?
- In the table 1 caption, it is written "same graph structure with training set" --> do you mean "same graph structure than the training set"?

---

> ### Author Response · Authors · 2017-12-19
> **Author response**
>
> Thank you for the constructive comment.
> We’ve posted a common response to the all reviewers as a separate comment above.
> We’d appreciate it if you go through the common response as well as this comment.
>
> Q1) Justification of why this direction of research is relevant
> A) We believe the first answer in the common response partially addresses this question. Even though options are pre-defined, the high-level planning problem itself is very challenging as discussed in the common response. We added Hierarchical Task Network (HTN) paragraph in section 2 to discuss how our problem is different from the planning literature. We also added new results in Section 5.6 that compare our method against a standard planning method (MCTS). We also show that our method can significantly improve MCTS by combining them together. We will motivate the problem better in the next revision.
>
>
> Q2) Future directions
> A) We appreciate you suggesting many interesting ways to extend our problem. We agree that it would be more interesting and challenging to have uncertainty in the task graph or to learn task graph or option itself. We are working on some of these directions. In this work, we focused on learning a generalizable agent which takes a richer and general form of task descriptions.
>
> Q3) Regarding the reward shaping used in RProp policy
> A) We clarify that the main idea of the RProp policy does not use any supervision and does not strongly benefit from human knowledge for the following reasons.
> 1) Compared to the usual reward shaping which often involves human-knowledge, our method “smoothes out” the reward information in the task graph in order to propagate reward information between related subtasks. Thus, the term “reward shaping” means “smoothing” here, and we removed “reward shaping” from the paper as it is confusing.
> 2) In fact, the agent always receives only the actual reward. The idea of the RProp policy is about how to form a “differentiable” task graph and how to backpropagate through it to get a reasonably good initial policy just from the task graph.
>
>
> Q4) Why NTS-scratch fails?
> A) Since this is related to the difficulty of the problem, please refer to the first answer in the common response. We found that even sophisticated search-based planning methods (e.g., MCTS) do not perform well compared to our method. Thus, it is not surprising that NTS-scratch fails to learn from scratch.
>
>
>
> Q5) Gamma with per-step penalty
> A) We agree that using gamma and per-step penalty together have a similar effect. However, many previous works [1-3] suggest that per-step penalty in addition to discount factor can be helpful for training especially in grid-world domain. For this reason, we used both discount factor and per-step penalty for better performance.
> [1] Konidaris et al., "Building Portable Options: Skill Transfer in Reinforcement Learning.”,  2007.
> [2] Melo et al., "Learning of coordination: Exploiting sparse interactions in multiagent systems.", 2009.
> [3] Konidaris et al., "Transfer in reinforcement learning via shared features.", 2012.
>
> Q6) Why does AND/OR operation take more than two input?
> A) For notational simplicity, we defined AND and OR operation which can take multiple input (not necessarily three) and are different from logical AND and OR operation. We added mathematical definitions at Appendix C.
>
> Q7) Why smoothed AND/OR function does not depend on the number of inputs?
> A) Both the AND/OR operation and smoothed AND/OR operation depend on the number of inputs as formulated in Appendix C. Please let us know if you need further clarification.
>
> Q8) Lack of detail of training
> A) We added more details at the Appendix B and D. In brief, we followed actor-critic framework without replay memory. Please let us know if you find any missing details.
>
> Q9) More experiment and examples in a different setting
> A) Can you please clarify what you mean by “type of tasks”? To our best knowledge, the prior work on hierarchical RL (e.g., HAM, MAXQ) and hierarchical planning (e.g., HTN) cannot directly address our problem as discussed in the common response above.
>
> Q10) Regarding episode length
> A) We found that the episode length that allows executing approximately 60-80% of total subtasks is not too long or too short so that the agent should consider both short-term and long-term dependencies between subtask to solve the problem. Note that remaining time is given as additional input to the agent so that the agent can perform different strategies depending on the time limit. The episode length is randomly sampled from a range according to the performance of the Random policy.
>
> Q11) Why wait until exactly 120 epochs before fine-tuning?
> A) NTS-RProp indeed converged earlier than 120 epochs. We wanted to make sure to wait until everything converges. We could stop earlier than 120 epochs as you suggested.

---

### Author Response · Authors · 2017-12-18
**Common response to all reviewers**

Dear reviewers,

Thank you for the valuable comments. We revised our paper according to your comments. So, we would appreciate if you take a look at the current revision.

We put a common response here as many of you raised similar questions/comments about the simplicity of the problem and the lack of comparison. (In addition, we provide individual responses to specific reviewers in separate comment sections.)

Q1) The proposed problem seems easy
A) The fundamental challenge in our problem is that the agent needs to generalize to new task dependency structures in testing time. To the best of our knowledge, there is no existing method (other than search-based methods) that can address this problem.

To further help the readers better understand how complex the problem is and how well our method performs, we performed additional experiments (Section 5.6), which demonstrate that even a sophisticated search method (MCTS) performs much worse than our method even with much larger amounts of search time budget (e.g., hundreds of simulated episodes instead of a single episode in our method).

In more detail, we also summarize several reasons why this problem is challenging as follows.
1) Our problem is essentially a combinatorial search problem where the optimal solution (i.e., optimal sequence of subtasks) cannot be computed in polynomial time. Given 15 subtasks, the number of valid solutions for each episode is approximately 600K. This is also the reason why we couldn’t scale up to a large number of subtasks and objects.
2) The agent should “infer” cost information from the observation. The task graph does not describe the cost of each subtask, and the agent should implicitly learn to predict the cost from the observation without any supervision.
3) Even without any dependencies between subtasks (no edges in the task graph), the agent should find the shortest path to execute subtasks, which becomes the infamous NP-hard Travelling Salesman Problem (TSP).
4) The agent needs to consider the time limit, because the optimal solution can be completely different depending on the time limit even with the same observation and task graph.


Q2) Lack of comparison to other hierarchical learning schemes (HAMs, options)
A) Please note that we do not claim novelty in proposing a new hierarchical learning scheme as our work is built on options framework and policy gradient methods. Instead, the main contribution of this work is 1) to propose a richer and general form of task descriptions (task graph) compared to the previous work on multi-task RL and 2) to propose a deep RL architecture for learning to optimally execute any arbitrary task graphs and observations. We will make this more clear in the next revision.

To our best knowledge, the prior work on hierarchical RL (e.g., HAM, MAXQ) cannot directly address our problem where the task description is given as a form of graph. For example, a HAM uses finite state machines to specify a partial policy. So, it is not straightforward to specify a general strategy for solving “any task graphs” using HAM, as our problem is essentially a combinatorial search problem. More importantly, such a partial policy should be hand-designed and heuristic.

An important thing to note is that most of the prior work on hierarchical RL considered a single-task policy (e.g., a fixed task in Taxi domain), whereas our problem is a multi-task learning problem where the agent should deal with many different tasks depending on the given task graph. This motivated us to propose a new architecture that is capable of executing many different and even unseen task graphs.


Q3) Domain is not standard
A) If we had aimed to propose a new hierarchical learning scheme, the evaluation could have been done using a standard domain. However, as discussed above, we aim to address a new problem: solving a combinatorial search problem without explicit search. Thus, we chose Mazebase environment which is a standard domain for evaluating multi-task policy [1-4] and flexible enough to implement our task graph execution problem.

We would also like to point out that the Taxi domain can be completely subsumed by the Mazebase domain and task graphs in our paper. For example, we can define 2 subtasks as follows: A (pick up passenger) and B (go to destination), and A is the precondition of B. Note that task graphs used in our experiment are much more complex than task dependencies in the Taxi domain.
[1] Sukhbaatar, et.al. (2016). Learning multiagent communication with backpropagation.
[2] Kulkarni, et.al. (2016). Deep successor reinforcement learning.
[3] Oh, et.al. (2017). Zero-shot task generalization with multi-task deep reinforcement learning.
[4] Thomas, et.al. (2017). Independently Controllable Features.


Q4) Details of training method
A) We added more details at the Appendix B and D. Please let us know if you find any missing details. We also plan to release the code to make the result reproducible.

---

### Decision · Program_Chairs · 2018-01-29
**ICLR 2018 Conference Acceptance Decision**

**Decision:**

Reject

**Comment:**

Paper presents and interesting new direction, but the evaluation leaves many questions open, and situation with respect to state of the art is lacking